# Antifungal Resistance in Clinical Isolates of *Aspergillus* spp.: When Local Epidemiology Breaks the Norm

**DOI:** 10.3390/jof5020041

**Published:** 2019-05-21

**Authors:** Mercedes Romero, Fernando Messina, Emmanuel Marin, Alicia Arechavala, Roxana Depardo, Laura Walker, Ricardo Negroni, Gabriela Santiso

**Affiliations:** Mycology Unit of the Infectious Diseases Hospital F.J. Muñiz, Reference Center of Mycology of Buenos Aires City, Buenos Aires C1282A, Argentina; fmessina35@gmail.com (F.M.); emamarin@hotmail.com (E.M.); aliarecha@hotmail.com (A.A.); roxanadepardo@yahoo.com.ar (R.D.); lgwalker2006@hotmail.com (L.W.); ricnegroni@hotmail.com (R.N.); santisog@hotmail.com (G.S.)

**Keywords:** Aspergilosis, azoles resistance, antifungal susceptibility, *Aspergillus*

## Abstract

Aspergillosis is a set of very frequent and widely distributed opportunistic diseases. Azoles are the first choice for most clinical forms. However, the distribution of azole-resistant strains is not well known around the world, especially in developing countries. The aim of our study was to determine the proportion of non-wild type strains among the clinical isolates of *Aspergillus* spp. To this end, the minimum inhibitory concentration of three azoles and amphotericin B (used occasionally in severe forms) was studied by broth microdilution. Unexpectedly, it was found that 8.1% of the isolates studied have a diminished susceptibility to itraconazole. This value turned out to be similar to the highest azole resistance rate reported in different countries across the world.

## 1. Introduction

Fungi of the genus *Aspergillus* have the capacity to cause diverse clinical pictures, from simple allergic forms to invasive aspergillosis, with a mortality rate higher than 80% in invasive forms when therapeutic measures are not applied quickly [1,2,3].

The aspergillosis clinical course depends on the patient´s underlying condition, the species involved, and the applied treatment. This makes it essential to identify the isolates and thus determine the antifungal susceptibility pattern to find the possible presence of intrinsic or acquired resistance to any drug [4,5]. Phylogenetic studies integrate data of partial DNA sequences to distinguish the different species within the section *Fumigati*, for instance. However, this approach is fastidious and not convenient for rapid identification in clinical practice [6].

Triazoles are usually the antifungals of choice for the treatment of different clinical forms of aspergillosis and in the prophylaxis of severe cases [7]. In long term treatment, particularly, the sustained use of these drugs might cause antimicrobial pressure responsible for selecting non-susceptible clones [8,9].

Resistance mechanisms are best studied in *A. fumigatus* and involve mutations in the *cyp51A* gene. Specific mutations can result in the lack of susceptibility to one or more triazoles. However, it is noteworthy that not all mutations contribute to azole resistance and resistance is not always caused by mutations in the *cyp51A* [10,11].

Acquired resistance to azoles would develop in response to the sustained exposure of fungi to azolic compounds. Based on this assumption, two possibilities have been proposed for its development: Exposure of the patient during treatment or in the environment where azoles for agriculture applications might be present, with consequent cross-resistance to azoles for clinical use [12,13].

Fungi reproduce massively in patients with aspergillosis, especially those with fungal balls in pre-formed cavities. In such cases, when submitted to prolonged treatment, sustained exposure to antifungals facilitates the emergence of resistance and the transmission of gene mutations to the new spores. Therefore, it is in this group of patients where the largest number of isolates with decreased susceptibility has been reported [14,15].

Given the increasing information in the last 25 years in relation to the appearance of isolates with decreased susceptibility to antifungals, the study of in vitro susceptibility is a fundamental tool to be able to opt for the correct treatment [3,16,17].

The Clinical and Laboratory Standards Institute (CLSI) has developed reference methods to evaluate the antifungal susceptibility of filamentous fungi and, in the absence of clinical breakpoints, the concept of epidemiological cut-off values (ECV) has been defined. The definition of an epidemiological cutoff value is the minimal inhibitory concentration (MIC) that separates a population into wild type and non-wild type isolates, considering the latter are those with possible acquired or mutational resistance based on their phenotypic MIC value.

An ECV is not a predictor of clinical success. The usefulness of an ECV lies in its ability to predict possible resistance to an antifungal agent, for which there is not enough data to establish breakpoints [18,19,20,21,22,23].

In spite of the fact that the reference method proposed by the CLSI to determine the MIC is laborious and is not habitually used in most clinical laboratories [24,25], in those centers which nurse a great number of aspergillosis cases it is recommended that this technique is performed to know the antifungal susceptibility pattern of their isolates and its resistance epidemiology [26,27].

The aim of our study was to evaluate the susceptibility profile to amphotericin B and the most relevant azoles of clinical isolates of *Aspergillus* spp. obtained from patients with aspergillosis or colonization of airways by these fungi.

## 2. Materials and Methods

A retrospective and descriptive study was carried out on 186 *Aspergillus* spp. isolates from different clinical episodes of 161 patients (one isolate per patient per episode). All the individuals were assisted at the Infectious Diseases Hospital Francisco J. Muñiz of Buenos Aires, Argentina, between 2012 and 2016. The clinical samples where the isolates came from were 6 sterile sites biopsies, 1 purulent collection, 37 bronchoalveolar lavages, and 142 sputa.

The vast majority of the isolates were obtained from patients with chronic forms of aspergillosis, especially intracavitary ones. This has a linear relationship with the large number of patients with tuberculosis backgrounds, who are nursed at our center. Pulmonary sequela, particularly cavities, constitute a fundamental risk factor for the mentioned type of aspergillosis.

The identification of the isolates was performed on the basis of the macro- and micro-morphological characteristics. Regarding the macroscopic analysis, characteristics of the colony, such as diameter, color of the front and back, texture and the presence of sclerotia, exudate drops, and diffusible pigment, were considered after growth on Czapek agar for 7 days at 28 °C. With regard to micro-morphology, characteristics, such as traits of the conidiophores, shape and diameter of the vesicles, arrangement of the methula or phialides on the vesicle and their length and width, diameter and color of the conidia, presence of Hülle cells, cleistothecia, and ascospores and their shape, size and color, were observed [28,29]. In the non-wild type *Aspergillus* section *Fumigati*, a temperature test (incubation at 48 °C) was performed to presumptively separate *A. fumigatus sensu stricto* from other cryptic species of the complex [6].

Isolates studied were 6 *Aspergillus* section *Terrei* (3.2%), 15 section *Flavi* (8.1%), 23 section *Nigri* (12.4%), and 142 section *Fumigati* (76.3%).

Antifungals of certified purity (itraconazole Panalab Argentina, voriconazole Pfizer UK, posaconazole MSD, and amphotericin B Oxoid) were used for the tests.

The susceptibility to itraconazole (ICZ), voriconazole (VCZ), and amphotericin B (AMB) was evaluated in all isolates, while posaconazole (PCZ) was tested in 76 of them (57 *Aspergillus* section *Fumigati*, 7 *Aspergillus* section *Nigri*, 6 *Aspergillus* section *Flavi* and 6 *Aspergillus* section *Terrei*).

The MICs of different antifungals were determined using the method of broth microdilution described in the M38-A2 document of the CLSI [18]. The assay was performed using Roswell Park Memorial Institute (RPMI) broth, inocula between 0.4 × 10^4^ to 5 × 10^4^ cfu/mL and incubation for 48 h at 35 °C. Two American Type Culture Collection (ATCC) control strains were used: *Paecilomyces variotii* and *Aspergillus fumigatus*. The MIC was defined as the minimum antifungal concentration capable of totally inhibiting growth.

To interpret the MICs obtained, epidemiological cut-off values shown in Table 1 were used.

ECVs capture ≥97.5% of the statistically modeled population.

When the MIC≤ECV isolates were considered as wild type, whilst when MIC>ECV decreased susceptibility was contemplated [21].

## 3. Results

The distribution of MIC values for different species and antifungals is presented in Figure 1.

The MIC_50_, MIC_90_, and MIC mode of the studied drugs are presented in Table 2.

Regarding the comparison of the MIC values obtained with the ECV for each species and antifungal, 15 isolates were obtained (14 section *Fumigati* and 1 section *Nigri*) with MIC values > ECV for itraconazol, so 8.1% of the isolates probably presented some kind of mechanism that reduced their susceptibility to this antifungal agent. Within the population of the 142 isolates of the section *Fumigati*, this percentage was set to 9.8%. All the non-wild type isolates from section *Fumigati* corresponded presumptively to *Aspergillus fumigatus sensu stricto* according to the temperature test results. The non-wild type *Aspergillus* section *Nigri* would need to undergo molecular analysis to be identified at species level. This was not performed during this study.

For voriconazole, three (1.6%) isolates had MIC values higher than ECV; all belonged to section *Fumigati*. These isolates had also shown decreased susceptibility to itraconazole, which suggests that cross-resistance was 20% in the group of isolates with reduced susceptibility to at least one azole.

In the case of amphotericin B and posaconazole, no isolates exhibited MIC values that exceeded the ECV.

## 4. Discussion

Since 1990 there have been reports about the acquired resistance of *Aspergillus* to azoles. Resistance data have increased significantly in recent years, especially for *A. fumigatus*. This has also been evident, but to a lesser extent, in amphotericin B [30,31]. Currently, there is serious concern about the emergence of resistance to itraconazole and the emergence of cross-resistance to multiple azoles. Even though its true incidence, its extent, and the impact on the clinical course of patients is not very clear [13,32], resistance could complicate their management and be associated with treatment failure [33].

Surveillance studies indicate that the prevalence of resistance varies widely among countries [34,35,36]. In some regions of the Netherlands, the percentage of patients infected with non-wild type *Aspergillus* to azoles reaches 12%. Resistance to azoles has also been observed, although sporadically, in France, India, Japan, China, Denmark, Switzerland, Norway, and Germany, but with prevalence values between 0% and 5% [11,33,35]. In Argentina, the prevalence of isolates with resistance or decreased susceptibility to antifungals is not known. The percentage of non-wild type strains for itraconazole susceptibility obtained in this study (8.1%) is comparable to those of the European countries with the highest proportion of non-wild type isolates.

The high percentage of isolates with reduced susceptibility to itraconazole observed is a fact of great importance, not only because it is the first information obtained from clinical isolates, but also because its magnitude reinforces the need for further studies for its use as a tool in the approach of patients with aspergillosis. It is necessary to carry out prospective analysis in order to understand the true implication of this phenomenon in the activity of this antifungal agent in vivo and in the clinical evolution of patients.

The results obtained show that somehow the local epidemiology breaks with the norms, especially for the chronic forms of aspergillosis. Paradoxically, an important percentage of strains turned out to be non-wild type for itraconazol, the most recommended antifungal and the most clinically used in our region. On the other hand, very low MIC values were found for posaconazole, which is the least recommended azole for this form of aspergillosis [35].

The fact that in the present study itraconazole is the antifungal agent for which a higher proportion of isolates has been found with decreased susceptibility inevitably leads us to think about its wide clinical use in our country, especially in patients with chronic forms that are treated in this center. Other authors have previously referred to the important contribution of non-wild type strains from patients with aspergilloma [36]. We hypothesize this would then be more related to a clinical exposure origin than to an environmental one. Molecular studies looking for specific mutations are imperative to confirm the origin of the non-wild type strains.

Another interesting issue that should be considered is the fact that developed countries of Europe have higher rates of resistance to voriconazole and posaconazole than the ones we found in our country. This indicates that there are important differences in the susceptibility profiles of strains from different places with their own environmental conditions and clinical management choices [30,37,38].

We have found very low MIC values for posaconazole (MIC 90 ≤ 0.03 μg/mL), perhaps due to its limited use.

In Argentina, newer azoles, such as isavuconazole and ravuconazole, are not yet available. This is the reason why these drugs have not been tested in this study [39].

The cases analyzed are too few to draw representative conclusions on cross-resistance.

To reach a true dimension of *Aspergillus* resistance, it is imperative to carry out susceptibility studies for all isolates from patients with aspergillosis, particularly in those with severe infections, to be able to consider an alternative therapy in geographical areas where there is a high prevalence of non-wild type isolates. It would also be useful to have epidemiological information about the region’s environmental isolates to infer the possible emergence of cross-resistance.

## Figures and Tables

**Figure 1 jof-05-00041-f001:**
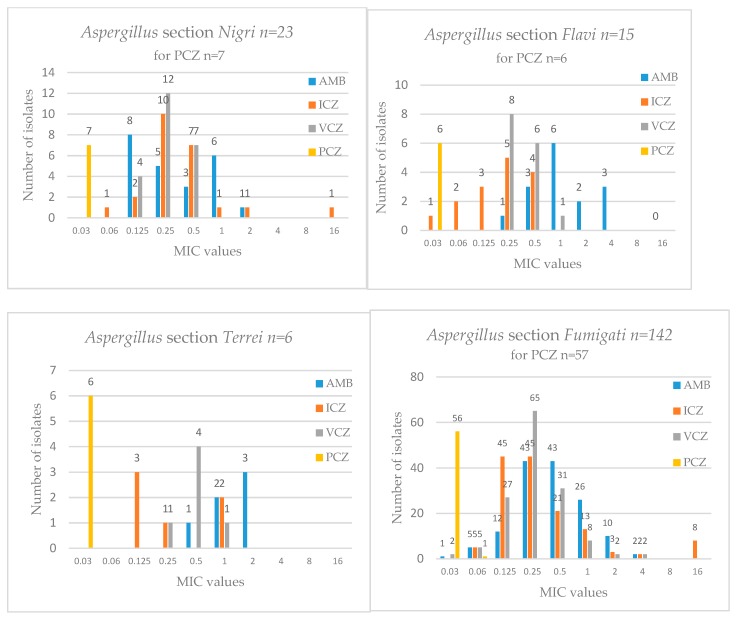
Number of isolates as a function of minimal inhibitory concentration (MIC) values for each drug and each species. AMB: Amphotericin B; ICZ: Itraconazole; VCZ: Voriconazole; PCZ: Posaconazole.

**Table 1 jof-05-00041-t001:** Cut-off values of the antifungals studied for the different *Aspergillus* species according to CLSI M59 2nd Ed. document.

Antifungal Agent	Species	Epidemiological Cut-off Value, μg/mL
Amphotericin B	*Aspergillus flavus*	4
	*Aspergillus fumigatus*	2
	*Aspergillus niger*	2
	*Aspergillus terreus*	4
Itraconazole	*Aspergillus flavus*	1
	*Aspergillus fumigatus*	1
	*Aspergillus niger*	4
	*Aspergillus terreus*	2
Voriconazole	*Aspergillus flavus*	2
	*Aspergillus fumigatus*	1
	*Aspergillus niger*	2
	*Aspergillus terreus*	2
Posaconazole	*Aspergillus flavus*	0.5
	*Aspergillus fumigatus*	0.5 *
	*Aspergillus niger*	2
	*Aspergillus terreus*	1

* (Reference [21]).

**Table 2 jof-05-00041-t002:** MIC50, MIC90, MIC mode and MIC range of the 4 antifungals tested for each species Section.

Species	Number of Isolates	Antifungals	MIC_90_ (μg/mL)	MIC_50_ (μg/mL)	MIC mode (μg/mL)	MIC Range (μg/mL)
*Aspergillus* section *Fumigati*						
	142	Itraconazole	0.5	0.25	0.25	0.06–16
	142	Voriconazole	0.5	0.25	0.25	0.03–8
	57	Posaconazole	0.03	0.03	0.03	0.03–0.06
	142	Anphotericin B	1	0.5	0.25	0.03–4
*Aspergillus* section *Nigri*						
	23	Itraconazole	1	0.25	0,25	0.06–16
	23	Voriconazole	0.5	0.25	0.25	0.125–0.5
	7	Posaconazole	*	*	0.03	0.03
	23	Anphotericin B	1	0.25	0.125	0.125–2
*Aspergillus* section *Flavi*						
	15	Itraconazole	0.5	0.25	0.25	0.03–0.5
	15	Voriconazole	0.5	0.25	0.25	0.25–1
	6	Posaconazole	*	*	0.03	0.03
	15	Anphotericin B	4	1	1	0.25–4
*Aspergillus* section *Terrei*						
	6	Itraconazole	*	*	0.125	0.125–1
	6	Voriconazole	*	*	0.5	0.25–1
	6	Posaconazole	*	*	0.03	0.03
	6	Anphotericin B	*	*	2	0.5–2

* MIC90 and MIC50 are not appropriate parameters when the number of tested isolates is less than 10.

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
