# Peer review of "Antifungal Resistance in Clinical Isolates of Aspergillus spp.: When Local Epidemiology Breaks the Norm"

_jof, 2019, doi:10.3390/jof5020041_

Reviewer 1 Report

The authors describe the rate of resistance in aspergillus species in a single center in Argentinia. Local epidemiological resistance data is important for clinicians, as diagnostic and treatment stratagies may be adjusted to this epidemiology. Therefore, the high resistance rate presented by the authors is a important finding. However, in aspergillus fumigatus it is very important to know which resistance mechanisms are found in your patients, as this affects diagnostic strategies. 

e.a. if resistance is due to "environmental" resistance mechanisms (like tr34/l98h and tr46/y121f/t28a and others), also patients not treated with azoles before should be checked for azole resistance. However, it is also possible that only cyp51a point mutations are found that are associated with long term therapy. In the latter, molecular assays that detect tr34/l98h and tr46 mutations are not usefull. Thus, it is very important to include molecular analysis of the resistant isolates. 

Major comments

1 as mentioned above, molecular analysis of resistance mechanism is of utmost importance 

2 It is important to know whether the isolates identified as aspergillus fumigatus complex are aspergillus fumigatus sensu strictu or cryptic species. A test that can be added is to analyse whether the isolate is able to grow at 48 degrees celcius. a. fumigatus can, but most cryptic species are not. Important to incorporate diagnostic and treatment stratagies. Some cryptic species are resistant for azoles. 

3 167-173. This paragraph is not true. In europe, most azole resistant isolates are resistant to itraconazole. MIC distributions are NOT significantly influenced by the use of either posaconazole/voriconazole/itraconazole. 

4 figures and tables should be changed: the cannot be interpretated as they lack info or are not well constructed. Some alternatives/options are mentioned below. 

Minor comments

5 25-26. Mortality in IA is not as high as 100% when treated. please correct to more realistic numbers. 

6 species names arei not written in italics throughout the manuscript. 

7 31-32 the authors mention tate that it is important to identify to the species level. see comment 2. 

8 67 wild population -> wild type population? 

9 75-77. in your population, but not in your environment. Maybe the resistant isolates are induced due to therapy and in your environment only WT isolates can be found? 

10 91-92. treatment of aspergillosis associated with intrapulmonary cavities is thought to be the primary source of treatmant associated resistance. (Howard - Frequency and Evolution of Azole

Resistance in Aspergillus fumigatus Associated with Treatment Failure). Consider mention this in the discussion. 

11 Table 1: ECV of a. niger seem very high. Might be true for a. niger complex including a. tubingensis? for a niger sensu strictu it should be much lower, for a. tubingensis is might be correct. 

12 121-122 CIM = MIC?

13 fig 1: the figure is not understandable. Consider making separate figures for a. fumigatus etc. and for posaconazole/voriconazole/itraconazole and L-AMB seperatly

14 FIG 1 CIM=MIC?

15 Table 2: for which species? only aspergillus funmgatus? all isolates? the table can not be interpretated without this info. 

16 143-144 reference 30 does not report resistance in L-AMB. 

17 165, clinical media? what do you mean?

18 176-179 not true for TR46/y121f/t289a. These isolates can be susceptible for itraconazole but resistant to voriconazole. see mic distrubutions from isolates from the Netherlands. 

Author Response

The authors describe the rate of resistance in aspergillus species in a single center in Argentinia. Local epidemiological resistance data is important for clinicians, as diagnostic and treatment stratagies may be adjusted to this epidemiology. Therefore, the high resistance rate presented by the authors is a important finding. However, in aspergillus fumigatus it is very important to know which resistance mechanisms are found in your patients, as this affects diagnostic strategies.

e.a. if resistance is due to "environmental" resistance mechanisms (like tr34/l98h and tr46/y121f/t28a and others), also patients not treated with azoles before should be checked for azole resistance. However, it is also possible that only cyp51a point mutations are found that are associated with long term therapy. In the latter, molecular assays that detect tr34/l98h and tr46 mutations are not usefull. Thus, it is very important to include molecular analysis of the resistant isolates.

Thank you so much for your comments and suggestions. We understand that molecular studies about the origin of resistance are imperative. Unfortunately, as we work in a public hospital (in an underdeveloped country) we don´t count with the resources or the funding to carry out this now.

The real aim of our paper was to show the reality we face every day in the routine work nursing patients. This is why traditional methods for the identification of isolates are used, to give you just an example. Patients or clinicians usually can´t be left waiting for molecular investigation (sending a sample to reference centers takes at least 20 days here), they need the sooner answer as possible about species (at least at Section level) and susceptibility.

We found that the percentage of non-wild type isolates is so high that is important to be reported in spite of not being sure about its origin. We guess the clinical exposure to azoles would be the most probable explanation as the vast majority of our patients have received long term therapy with those drugs. I hope in a near future we will be able to perform these studies to produce a more complete report of the situation in our country.

Major comments

1 as mentioned above, molecular analysis of resistance mechanism is of utmost importance

As mentioned above it is not possible for us to perform this right now. I have added a comment about the former and about the confirmation of the origin of resistance in the discussion.

2 It is important to know whether the isolates identified as aspergillus fumigatus complex are aspergillus fumigatus sensu strictu or cryptic species. A test that can be added is to analyse whether the isolate is able to grow at 48 degrees celcius. a. fumigatus can, but most cryptic species are not. Important to incorporate diagnostic and treatment stratagies. Some cryptic species are resistant for azoles.

We have done the test you suggested. All the non-wild type strains from the Section Fumigati were able to grow at 48 degrees. This information was added to the manuscript

3 167-173. This paragraph is not true. In europe, most azole resistant isolates are resistant to itraconazole. MIC distributions are NOT significantly influenced by the use of either posaconazole/voriconazole/itraconazole.

The paragraph was corrected:

Another interesting issue is that developed countries of Europe have higher rates of resistance to voriconazole and posaconazole than the ones we found in our country. This indicates that there are important differences in the susceptibility profiles of strains from different places with their own environmental conditions and clinical management choices [37].

4 figures and tables should be changed: the cannot be interpretated as they lack info or are not well constructed. Some alternatives/options are mentioned below.

Tables and figures were corrected.

Minor comments

5 25-26. Mortality in IA is not as high as 100% when treated. please correct to more realistic numbers.

We corrected the number and updated references.

6 species names arei not written in italics throughout the manuscript.

All species names are now in italics 

7 31-32 the authors mention tate that it is important to identify to the species level. see comment 2.

That was changed for:  to species level when possible or at least to Section level.

8 67 wild population -> wild type population?

Wild population was changed for wild type population

9 75-77. in your population, but not in your environment. Maybe the resistant isolates are induced due to therapy and in your environment only WT isolates can be found?

The Word “Enviroment” was wrongly used in that phrase, maybe there is not an exact translation for what we mean. It was now changed for “population”.

10 91-92. treatment of aspergillosis associated with intrapulmonary cavities is thought to be the primary source of treatmant associated resistance. (Howard - Frequency and Evolution of Azole

Resistance in Aspergillus fumigatus Associated with Treatment Failure). Consider mention this in the discussion.

I have mentioned this in the discussion and I also added the reference you suggested.

11 Table 1: ECV of a. niger seem very high. Might be true for a. niger complex including a. tubingensis? for a niger sensu strictu it should be much lower, for a. tubingensis is might be correct.

That piece of information was extracted from the second edition of the M59 document of the CLSI. (2018)

That data is referred to A. niger in the mentioned document.

12 121-122 CIM = MIC?

Yes, that was corrected by now.

13 fig 1: the figure is not understandable. Consider making separate figures for a. fumigatus etc. and for posaconazole/voriconazole/itraconazole and L-AMB seperatly

Separated figures are included by now.

14 FIG 1 CIM=MIC?

Yes, that was corrected.

15 Table 2: for which species? only aspergillus funmgatus? all isolates? the table can not be interpretated without this info.

The table was remade including MIC50, MIC90 and MIC range separately  for each species

16 143-144 reference 30 does not report resistance in L-AMB.

A new reference was added

17 165, clinical media? what do you mean?

That was a bad translation, it was replaced for “its wide clinical use in our country”

18 176-179 not true for TR46/y121f/t289a. These isolates can be susceptible for itraconazole but resistant to voriconazole. see mic distrubutions from isolates from the Netherlands.

You are right, that phrase was deleted.

Finding strains susceptible to itraconazole and resistant to voriconazole is very rare here, that led me to that biased conclusion.

Reviewer 2 Report

In this study, authors evaluated MIC of 186 Aspergillus clinical isolates from Argentina, against azoles drugs and amphotericin B. They claimed 8,1% of the isolates with a MIC > ECV for itraconazole and 1,6% of the isolates with a MIC > ECV for voriconazole.

Major comment:

Isolates were identified using conventional methods, thus the isolates were classified at the section level. However, authors applied ECV of species leading to confused results. To confirm the resuts, isolates with MIC > ECV must be identified at species level, using beta-tubulin. 

Minor comment:

Figure 1 should be separeted into 4 figures for each section because ECV are differents.

It would be interesting to explore mechanism of resistance implying CYP51A gene.

Lines 176-179: this sentence should be moderated because some resistant isolats carying TR46/Y121F/T289A alteration from environement, show resistance to voriconazole but sensibility to itraconazole. 

How did evolve MIC through years? did you find more MIC > ECV in 2016 than in 2012?

Author Response

In this study, authors evaluated MIC of 186 Aspergillus clinical isolates from Argentina, against azoles drugs and amphotericin B. They claimed 8,1% of the isolates with a MIC > ECV for itraconazole and 1,6% of the isolates with a MIC > ECV for voriconazole.

Major comment:

Isolates were identified using conventional methods, thus the isolates were classified at the section level. However, authors applied ECV of species leading to confused results. To confirm the resuts, isolates with MIC > ECV must be identified at species level, using beta-tubulin.

I understand that species level identification is important but unfortunately as we work in a public hospital  we don´t have the resources or funding to perform molecular studies on these strains now. I have carried out a temperature test (incubation at 48 degrees) to presumptively separate Aspergillus fumigatus sensu stricto from the cryptic species. This information has already been included in the manuscript.

Minor comment:

Figure 1 should be separeted into 4 figures for each section because ECV are differents.

Tables and figures were corrected and separated for each species.

It would be interesting to explore mechanism of resistance implying CYP51A gene.

 We understand that molecular studies about the origin of resistance are imperative. Unfortunately, as we work in a public hospital (in an underdeveloped country) we don´t count with the resources or the funding to carry out this now.

Our paper has tried to show the reality we face every day in the routine work nursing patients, this is why traditional methods for the identification of isolates are used, just to give you an example. Patients or clinicians usually can´t be left waiting for molecular investigation (sending a sample to reference centers takes at least 20 days here), they need the sooner answer as possible about species (at least at Section level) and susceptibility.

We found that the percentage of non-wild type isolates is so high that is important to be reported in spite of not being sure about its origin. I hope in a near future we will be able to perform these studies to produce a more complete report of the situation in our country.

Lines 176-179: this sentence should be moderated because some resistant isolats carying TR46/Y121F/T289A alteration from environement, show resistance to voriconazole but sensibility to itraconazole.

You are right, that phrase was deleted.

Finding strains susceptible to itraconazole and resistant to voriconazole is very rare here, that lead us to that biased conclusion.

How did evolve MIC through years? did you find more MIC > ECV in 2016 than in 2012?

The percentage of non-wild type isolates has remained constant throughout these years.

Reviewer 3 Report

This ms provides a look at the azole resistance in Aspergillus spp in Argentina. The major issues that I have with the paper is that the authors tend to use ECVs as clinical breakpoints and that they do not employ the most recent CLSI guidelines. In addition they fail to address one of the major points in the title "When local epidemiology breaks the norm". This theme needs to be addressed in the discussion.

Specific comments

The English must be improved. A specific example is seen in the paragraph encompassed in lines 63-69. In addition to unclear wording the definition of ECV is not exactly accurate. The authors should consult the CLSI doc M57Ed1 Principles and Procedures for the Development of Epidemiological Cutoff Values for Antifungal Susceptibility Testing, 1st Edition.pdf for a clear definition.

Specify that 1 isolate per patient infection episode was included in the study.

Several refs are out of date including ref 1 and refs 18 (M38Ed3E Reference Method for Broth Dilution Antifungal Susceptibility Testing of Filamentous Fungi, 3rd Edition.pdf), 19 (M61Ed1E Performance Standards for Antifungal Susceptibility Testing of Filamentous Fungi, 1st Edition.pdf) and 21(M57Ed1 Principles and Procedures for the Development of Epidemiological Cutoff Values for Antifungal Susceptibility Testing, 1st Edition.pdf; M59Ed2E Epidemiological Cutoff Values for Antifungal Susceptibility Testing, 2nd Edition.pdf). Regarding ref 19 as stated-disk diffusion testing was not performed so not sure if that is necessary at all.

It should be noted that whereas azoles are fungistatic vs yeasts they are often fungicidal vs moulds. Also please use the terms wild-type and non-wild-type rather than susceptible or resistant when applying ECVs.

Presently the state-of-the-art for fungal ID is either MALDI (yeasts ) or sequence-based (yeasts and moulds).

The lack of characterization of resistance mechanisms is a limitation.

It should be noted that an agar screen method using a fixed concentration of itraconazole is often used in studies such as this.

Author Response

This ms provides a look at the azole resistance in Aspergillus spp in Argentina. The major issues that I have with the paper is that the authors tend to use ECVs as clinical breakpoints and that they do not employ the most recent CLSI guidelines. In addition they fail to address one of the major points in the title "When local epidemiology breaks the norm". This theme needs to be addressed in the discussion.

Thank you so much for your comments and suggestions.

There was a mistake in the references. CLSI updated documents were used along the study but as the tests were performed through many years, documents have changed many times during this period. The first documents that were used were wrongly included in the references.

References are updated now.

Specific comments

The English must be improved. A specific example is seen in the paragraph encompassed in lines 63-69. In addition to unclear wording the definition of ECV is not exactly accurate. The authors should consult the CLSI doc M57Ed1 Principles and Procedures for the Development of Epidemiological Cutoff Values for Antifungal Susceptibility Testing, 1st Edition.pdf for a clear definition.

English was revised by an English language professor. The definition of ECV was corrected.

Specify that 1 isolate per patient infection episode was included in the study.

That issue was clarified.

Several refs are out of date including ref 1 and refs 18 (M38Ed3E Reference Method for Broth Dilution Antifungal Susceptibility Testing of Filamentous Fungi, 3rd Edition.pdf), 19 (M61Ed1E Performance Standards for Antifungal Susceptibility Testing of Filamentous Fungi, 1st Edition.pdf) and 21(M57Ed1 Principles and Procedures for the Development of Epidemiological Cutoff Values for Antifungal Susceptibility Testing, 1st Edition.pdf; M59Ed2E Epidemiological Cutoff Values for Antifungal Susceptibility Testing, 2nd Edition.pdf). Regarding ref 19 as stated-disk diffusion testing was not performed so not sure if that is necessary at all.

References were updated.

It should be noted that whereas azoles are fungistatic vs yeasts they are often fungicidal vs moulds. Also please use the terms wild-type and non-wild-type rather than susceptible or resistant when applying ECVs.

The terms were replaced and the fungistatic wrong concept deleted.

Presently the state-of-the-art for fungal ID is either MALDI (yeasts ) or sequence-based (yeasts and moulds).

I understand that species level identification is important but as we work in a public hospital (in an undeveloped country) unfortunately we don´t have the resources or funding to perform molecular studies on this strains now. I have carried out a temperature test (incubation at 48 degrees) to presumtively separate Aspergillus fumigatus sensu stricto from the cryptic species. This information was included in the manuscript by now.

The lack of characterization of resistance mechanisms is a limitation.

I know that molecular studies about the origin of resistance are imperative. Unfortunately, we don´t count with the resources or the funding to carry out this now.

Our paper has tried to show  the reality we face every day in the routine work nursing patients, this is why traditional methods for the identification of isolates are used, just to give you an example. Patients or clinicians usually can´t be left waiting for molecular investigation (sending a sample to reference centers take at least 20 days here), they need the sooner answer as posible about species (at least at Section level) and susceptibility.

We found that the percentage of non-wild type isolates is so high that is important to be reported in spite of not being sure about their origin. I hope in a near future we will be able perform these studies to produce a more complete report of the situation in our country.

It should be noted that an agar screen method using a fixed concentration of itraconazole is often used in studies such as this.

Thank you so much for your comments and suggestions

Reviewer 4 Report

The manuscript submitted by Romero and colleagues reports an analysis of the pattern of azole-resistance among clinical Aspergillus spp. collected in the greater city of Buenos Aires, Argentina. By broth microdilution and CLSI epidemiological cut-off values (ECV), 8.1% of the isolates studied had decreased susceptibility to itraconazole, the triazole that is most frequently used in this region of the world. Decreased susceptibility to posaconazole and voriconazole was much less frequent. Molecular analyses of the genetic background of observed resistance were not performed.

Altogether, this is an interesting and well-structured report. I have the following comments and /or suggestions:

1.      The manuscript needs the attention of the editorial staff regarding English syntax – the meaning of several sentences and passages in the text is difficult to comprehend.

2.      The introduction is too long and needs to be shortened substantially – the authors are losing the reader on the way to the actual study. The manuscript will be published in a mycological journal so that most of the background is perfectly known by any reader.

3.      Introduction, line 42: The azoles are not strictly fungistatic but display organism-dependent pharmacodynamics in vitro. Pl. consider.

4.      Introduction, line 52: I am not aware that formation of a fungus ball and sporulation is the biologic explanation for secondary resistance in chronic aspergillosis.

5.      Introduction, line 63: Pl. note that the ECV may be suitable to define susceptibility, but it does not allow for any assumptions on clinical efficacy as it is an arbitrary endpoint selected in a situation with persistent lack of pharmacokinetic/pharmacodynamic correlations that would involve the MIC.

6.      Introduction, line 83: Pl. be clear to what the term environment refers to – isolates from the environment or the medical setting in and around your hospital.

7.      Methods, line 95: For a precise analysis that would be comparable to data from other regions, a molecular specification of the isolates would have been desirable.

8.      Methods, line 112: Pl. provide information on whether control strains were used when you did the broth microdilution analyses.

9.      Methods, line 121 and figure 1: Pl. replace the term CIM by the term MIC.

10.  Results, line 134: Molecular analysis of the isolates with elevated MIC values to antifungal triazoles would have been desirable for comparison (pl. see point 7).

11.  Discussion, line 167: The association of clinical use of an antifungal triazole with emergence of resistance in patients is speculative and a hypothesis at this point – pl. consider.

Author Response

The manuscript submitted by Romero and colleagues reports an analysis of the pattern of azole-resistance among clinical Aspergillus spp. collected in the greater city of Buenos Aires, Argentina. By broth microdilution and CLSI epidemiological cut-off values (ECV), 8.1% of the isolates studied had decreased susceptibility to itraconazole, the triazole that is most frequently used in this region of the world. Decreased susceptibility to posaconazole and voriconazole was much less frequent. Molecular analyses of the genetic background of observed resistance were not performed.

Thank you so much for your comments and suggestions. We understand that molecular studies about the origin of resistance are imperative. Unfortunately, as we work in a public hospital (in an underdeveloped country) we don´t count with the resources or the funding to carry out this now.

Our paper has tried to show the reality we face every day in the routine work nursing patients, this is why traditional methods for the identification of isolates are used, just to give you an example. Patients or clinicians usually can´t be left waiting for molecular investigation (sending a sample to reference centers take at least 20 days here), they need the sooner answer as posible about species (at least at Section level) and susceptibility.

We found that the percentage of non-wild type isolates is so high that is important to be reported in spite of not being sure about its origin. I hope in a near future we can perform these studies to produce a more complete report of the situation in our country.

Altogether, this is an interesting and well-structured report. I have the following comments and /or suggestions:

1.      The manuscript needs the attention of the editorial staff regarding English syntax – the meaning of several sentences and passages in the text is difficult to comprehend.

The manuscript has been revised by an English language professor by now.

2.      The introduction is too long and needs to be shortened substantially – the authors are losing the reader on the way to the actual study. The manuscript will be published in a mycological journal so that most of the background is perfectly known by any reader.

The introduction was not substantially reduced as it contains many changes other revisors asked for.

3.      Introduction, line 42: The azoles are not strictly fungistatic but display organism-dependent pharmacodynamics in vitro. Pl. consider.

This was corrected.

4.      Introduction, line 52: I am not aware that formation of a fungus ball and sporulation is the biologic explanation for secondary resistance in chronic aspergillosis.

Some comments about this and a new reference about the importance of aspergilloma clinical picture in the occurrence of resistance was added, as another revisor suggested.

5.      Introduction, line 63: Pl. note that the ECV may be suitable to define susceptibility, but it does not allow for any assumptions on clinical efficacy as it is an arbitrary endpoint selected in a situation with persistent lack of pharmacokinetic/pharmacodynamic correlations that would involve the MIC.

The concept of ECV was reformulated

6.      Introduction, line 83: Pl. be clear to what the term environment refers to – isolates from the environment or the medical setting in and around your hospital.

The word “environment” in that line was referred to the population in and around our place. It was a wrong translation that has now been corrected

7.      Methods, line 95: For a precise analysis that would be comparable to data from other regions, a molecular specification of the isolates would have been desirable.

As mentioned before we don´t have now the possibility to run molecular tests. I have included a temperature test (incubation at 48 degrees) at least to presumptively separate A. fumigatus sensu stricto from the cryptic species of the Section.

8.      Methods, line 112: Pl. provide information on whether control strains were used when you did the broth microdilution analyses.

P. variottii ATCC MYA-3630 and Aspergillus fumigatus ATCC 204305 were used. This information was added.

9.      Methods, line 121 and figure 1: Pl. replace the term CIM by the term MIC.

Replaced.

10.  Results, line 134: Molecular analysis of the isolates with elevated MIC values to antifungal triazoles would have been desirable for comparison (pl. see point 7).

Due to the lack of resources and/or funding we are not able to do this now. I hope this is not a restriction in the acceptance of my manuscript. I hope we will manage to do this in the near future.

11.  Discussion, line 167: The association of clinical use of an antifungal triazole with emergence of resistance in patients is speculative and a hypothesis at this point – pl. consider.

That was clarified as a hypothesis. It was also mentioned that molecular studies are necessary to know the real origin of resistance.

Round  2

Reviewer 1 Report

The authors changed most of the proposed modifications.The authors explain why the did not include molecular analysis: molecular analysis is not possible due to limited resources.

1 Change CIM to MIC in figures. 

Author Response

Thank you so much for your corrections and suggestions.

The figs. are corrected by now.

English has been re-checked.

The Introduction has been reduced as other revisor asked for.

Kind regards

Reviewer 2 Report

The manuscript has been improved with a better description of methodology than in the first version. I have only minor comments:

lines 62-63: "clinical breakpoints" instead of "clinical cut-off points"

Add a sentence about non-wild type Aspergillus section Nigri  which needs species molecular identification.

Author Response

Thank you so much for your corrections and suggestions.

"Clinical Cut off points" has been replaced for clinical breakpoints as you suggested.

A sentence about typing non-wild type Aspergillus section nigri has been included.

English has been re-checked.

The introduction has been reduced as other revisor asked for.

Kind regards

Reviewer 3 Report

Revised as suggested

Much improved

Please change CIM to MIC in the Figs

Please note that MIC90 is not appropriate when the number of isolates tested is less than 10

Author Response

Thank you so much for your suggestions and corrections.

Figs has been corrected.

MIC90 and MIC50 values have been deleted when less than 10 isolates were tested. An explanation was included under the table.

English was re-checked

The introduction was reduced as other revisor asked it for

Kind regards

Reviewer 4 Report

Starting with the abstract (last sentence), the manuscript still needs the attention of the editorial staff regarding English syntax.

The introduction is still way too long and reads like the introduction to a doctoraal theses and contains points that may be reserved for the discussion – the authors should consider that they submit their manuscript to the Journal of Fungi that is being read by experts in the field.

Author Response

Thank you so much for your comments.

English has been re-checked.

The introduction has been reduced by now.

Kind regards